# Endoplasmic Reticulum Stress and Unfolded Protein Response in Breast Cancer: The Balance between Apoptosis and Autophagy and Its Role in Drug Resistance

**DOI:** 10.3390/ijms20040857

**Published:** 2019-02-16

**Authors:** Lorenza Sisinni, Michele Pietrafesa, Silvia Lepore, Francesca Maddalena, Valentina Condelli, Franca Esposito, Matteo Landriscina

**Affiliations:** 1Laboratory of Pre-Clinical and Translational Research, IRCCS, Referral Cancer Center of Basilicata, 85028 Rionero in Vulture, Italy; lorisi@hotmail.com (L.S.); michele.pietrafesa@crob.it (M.P.); silvia.lepore@crob.it (S.L.); francesca.maddalena@crob.it (F.M.); valentina.condelli@crob.it (V.C.); 2Department of Molecular Medicine and Medical Biotechnology, University of Napoli Federico II, 80131 Naples, Italy; franca.esposito@unina.it; 3Medical Oncology Unit, Department of Medical and Surgical Sciences, University of Foggia, 71100 Foggia, Italy

**Keywords:** endoplasmic reticulum stress, unfolded protein response, breast cancer, apoptosis, autophagy, drug resistance, hormone therapy

## Abstract

The unfolded protein response (UPR) is a stress response activated by the accumulation of unfolded or misfolded proteins in the lumen of the endoplasmic reticulum (ER) and its uncontrolled activation is mechanistically responsible for several human pathologies, including metabolic, neurodegenerative, and inflammatory diseases, and cancer. Indeed, ER stress and the downstream UPR activation lead to changes in the levels and activities of key regulators of cell survival and autophagy and this is physiologically finalized to restore metabolic homeostasis with the integration of pro-death or/and pro-survival signals. By contrast, the chronic activation of UPR in cancer cells is widely considered a mechanism of tumor progression. In this review, we focus on the relationship between ER stress, apoptosis, and autophagy in human breast cancer and the interplay between the activation of UPR and resistance to anticancer therapies with the aim to disclose novel therapeutic scenarios. The hypothesis that autophagy and UPR may provide novel molecular targets in human malignancies is discussed.

## 1. Introduction

The endoplasmic reticulum (ER) is an intracellular organelle that contributes to the production and folding of cellular proteins and is involved in the maintenance of cellular homeostasis and the subtle balance between health and disease [1]. The ER is responsible for synthesis, maturation, folding, quality control, and degradation of secreted and transmembrane proteins, and guarantees that only correctly folded proteins can reach their cell compartment [1]. A specific ER stress pathway is activated when unfolded or misfolded proteins accumulate within the ER lumen [1], known as the unfolded protein response (UPR). Wrongly folded proteins can accumulate when ER protein folding capacity is overwhelmed by cellular demand and/or cellular energy availability is not sufficient to correctly fold proteins synthesized into the ER [1]. The UPR helps cells to restore homeostasis using different mechanisms: (i) by attenuating protein synthesis, (ii) by increasing the capacity of the ER to fold proteins and clear unfolded/misfolded proteins [2], and (iii) by activating chaperones/heat shock proteins to assist toward misfolded protein accumulation and cell cycle arrest [2].

The UPR leads to changes in activities of key regulators, integrating pro-death and pro-survival signals and functions, thus determining cell fate. This process is driven by signals that crosstalk among plasma membrane, ER, cytosol, mitochondria, and nucleus, leading to changes in cellular metabolism necessary to enable induction or repression of apoptosis and/or autophagy. The pathologic activation of the UPR is involved in the pathogenesis of several human diseases: cell death initiation has implications mainly in metabolic, neurodegenerative, and inflammatory diseases [3], whereas survival signals are relevant in malignancies [4]. In breast tumors, stress arises from hypoxia and nutrient deprivation induced by cytotoxic and endocrine therapeutic interventions and several lines of evidence suggest that chronic activation of the UPR is associated with therapy resistance and disease recurrence [5,6]. Moreover, several studies linked estrogen receptor signaling to the regulation of the UPR [7,8]: the glucose regulated protein 78 (GRP78), also called binding immunoglobulin protein (BiP), and the x-box binding protein 1 (XBP1) are upregulated in endocrine- and chemotherapy-resistant breast cancers [7,9,10,11,12,13]. In such a context, molecular mechanisms leading to chronic activation of UPR are currently viewed as novel potential targets to restore drug sensitivity.

## 2. ER Function: From Protein Production to UPR

The mail function of ER is to ensure an efficient quality control to guarantee that only properly folded proteins can reach their final destination [14]. This maturation process is finely regulated by specific chaperones of the reticular compartment. They can be gathered into three groups: GRP78 and GRP94 that facilitate the assembly and folding of unspecific proteins [15]; calnexin and calreticulin, lectins involved in the maturation process of glycoproteins [16]; and Heat shock protein 47 (Hsp47) specifically for collagen [17]. Furthermore, there is another functional class of enzymes that catalyzes the formation and/or disruption of disulfide bonds, such as protein disulfide isomerases (PDIs) [15]. Under physiological conditions, damaged or incorrectly folded proteins are sequestered and eliminated through a process called endoplasmic reticulum-associated protein degradation (ERAD) [18]. Calcium has a central role in this process, since reticular chaperones have different values of affinity for calcium and their activity is modulated by fluctuations in the concentration of this ion [19]. Moreover, the folding process and any refolding occur with a net energy consumption in the form of ATP, so any event that involves a modification of intracellular energy levels can block their activity. The activity of PDIs involves a net production of ROS, and when the capacity of the reticular antioxidant systems is saturated, the reticular homeostasis is compromised [20]. These stress stimuli are able to undermine the correct function of protein complexes used to guarantee the correct folding of newly synthesized proteins with consequential aggregation and protein misfolding, a condition better defined as “reticular stress” [19,20]. In tumors, the rapid and uncontrolled cell growth can alter reticular homeostasis due to intrinsic factors, as the increased rate of synthesis of oncoproteins, and extrinsic factors, such as nutrients and oxygen deprivation, which results in the establishment of a hypoxic state, acidosis, and starvation [14]. In order to struggle “protein-toxic” stress, cells respond by activating a signaling pathway called UPR that results in the reprogramming of a series of events that can be pro-survival or pro-death, depending on the extent of the damage or the length of the stress [21].

### 2.1. UPR Signaling

Three major sensors control the UPR: the inositol requiring enzyme 1 (IRE1), the protein kinase RNA-activated (PKR)-like ER kinase (PERK), and the activating transcription factor 6 (ATF6). These stress sensors are bound by the ER chaperone, GRP78/BiP, and are maintained in an inactive state [22,23] (Figure 1). Misfolded proteins, stored in the ER, activate GRP78/BiP, induce the expression of ER-resident chaperones, and transiently decrease protein synthesis. The UPR rebalances protein load and folding, thus restoring ER capacity, and for this reason, it is considered an adaptive and cytoprotective process [24]. The starting signal for UPR is the activation and homo-dimerization of PERK and IRE1, subsequently to the trans-autophosphorylation of their cytoplasmic components, whereas the activation of ATF6 occurs with its translocation to the Golgi apparatus (Figure 1). GPR78 has a crucial role in cell protection against ER stress: it is associated with the luminal domains of the UPR transducers, preventing PERK and IRE1 homo-dimerization, and impeding the translocation of ATF6 to the Golgi. When misfolded proteins increase in the ER lumen, GRP78 is recruited for protein folding, prevents Ca^2+^ release into the cytosol, thus inhibiting the cell death cascade [25], and releases UPR sensors. Upon release by GRP78, ATF6 is cleaved in the Golgi apparatus, translocates to the nucleus, and induces the expression of genes coding for enzymes responsible of protein folding and ER-resident molecular chaperones [26,27,28], including *GRP78*, among others [29] (Figure 1).

PERK activation induces the serine-phosphorylation of eIF2α and inhibits the translational activity of eIF2B, with the consequent block of protein synthesis [30]. During ER stress, this process reduces the protein overload and the accumulation of misfolded proteins, allowing the synthesis of Cap-independent transcripts [31] (Figure 1).

The third regulator of UPR is IRE1, whose endoribonuclease activity is responsible for the production of a splicing variant of XBP1, called XBP1s [32]. This transcriptional factor alleviates the ER stress through the activation of a series of downstream genes involved in protein secretion, maturation and degradation [32] (Figure 1).

Misfolded and aberrant proteins are degraded using a systemic pathway: detrimental proteins are tagged by chaperones, poly-ubiquitinated, and degraded in the 26S subunit of the proteasome [33], directing the cell towards adaptive UPR [34]. Afterwards, if the cell is able to overcome the ER stress condition, a series of pro-survival events are activated, whereas if the stress condition persists, UPR undertakes the pro-death pathway. It is important to note that several human cancers (i.e., hepatocellular, lung, pancreatic, and breast cancer) are characterized by prolonged and uncontrolled activation of UPR and this promotes tumor growth and therapy resistance [35]. Thus, UPR is presently evaluated as a molecular target for therapeutic interventions aimed at modulating UPR as a cancer-killing strategy.

### 2.2. The Unfolded Protein Response and the Balance between Autophagy, Survival, and Apoptosis

The UPR regulates different pathways to restore metabolic homeostasis, blocking or promoting cell death based on its capacity to remove the stress condition or not. This adaptive process involves the regulation and the reciprocal integration of autophagy and apoptosis. Indeed, autophagy is also associated to both cell survival and death [36]: pro-death signaling is usually activated to eliminate cells deprived of key proteins or altered by incorrect secretion of hormones, growth factors, or accumulation of misfolded proteins, or subjected to DNA damage or oxidative stress [37]. This is a natural process through which cells recycle organelles and damaged or not necessary proteins [38]. The process starts with (i) the formation of double membrane structures which phagocytize targeted molecules, (ii) the appearance of cytoplasmic vacuoles, and (iii) the increased cleavage of microtubule-associated protein 1 light chain 3 (LC3), following its transcription by PERK/eIF2α/ATF4 axis [39] (Figure 2). In physiological conditions, basal autophagy removes damaged organelles or old proteins, releasing into the cytosol degradation products coming from specific organelles, easily identifiable as mitochondria (mitophagy), ribosomes (ribophagy), and ER (reticulophagy) [39]. Similar to UPR, autophagy is associated to cell survival [40], being finalized to correct the energy imbalance, establish correct protein folding, and recycle cellular contents [41]. However, the persistence of the autophagic process is no longer maintained for survival; it could evolve in apoptotic or autophagic cell death and this may reflect the necessity to eliminate cells with altered key proteins or subjected to oxidative stress and DNA damage [42].

Two signaling pathways are responsible of the apoptotic cascade: the intrinsic or mitochondrial pathway and the extrinsic or death receptor pathway. During the activation of the intrinsic pathway, apoptotic stimuli induce changes in the mitochondrial inner membrane potential causing the exposure of pro-apoptotic proteins [43]. Ca^2+^ is released from the ER into the cytosol and activates caspase-12; PERK phosphorylates the alpha subunit of eIF2α and this in turn activates ATF4, which acts on the transcription factor CCAAT-enhancer-binding protein homologous protein (*CHOP*) promoter [44] (Figure 1). These factors stimulate the expression of pro-apoptotic proteins with the activation of DNA damage inducible protein 34 (GADD34), leading to an increase in protein synthesis [45,46], the downregulation of antiapoptotic proteins, and induce the release of cytochrome c into the cytosol activating caspase-3 and caspase-9 signaling [47,48]. In this scenario, it should be emphasized that PERK and ATF4 are fundamentally pro-survival pathways, but they can also promote apoptosis through the activation of CHOP, the main mediator of apoptosis induced by the UPR [49,50].

Different is the molecular mechanism leading to activation of the extrinsic pathway. Pro-death extracellular stimuli activate members of the tumor necrosis factor receptor superfamily and this leads to induction of the apoptotic cascade [51]. However, intracellular stimuli, during the ER stress process, are also responsible for the induction of the extrinsic pathway through a specific activation of c-Jun N-terminal kinase (JNK) [52]. These signals involve primarily IRE1 pathway, which (i) overcomes the transcriptional blockade induced by PERK [53], (ii) digests specific miRNAs through its endoribonuclease domain (IRE1-dependent decay of mRNA, RIDD) [54], and iii) recruits tumor necrosis factor receptor-associated 2 (TRAF2) with downstream activation of Apoptosis signal-regulating kinase 1 (ASK1), JNK [52], and p38 MAPK pathways [55] (Figure 3). JNK phosphorylation regulates the expression of specific Bcl-2 family members, favoring the activation of proapoptotic genes and the parallel downregulation of antiapoptotic proteins [56,57]. On one hand, the p38 MAPK pathway promotes the phosphorylation of CHOP serine residues 78 and 81, increasing its transcriptional activity [58], and this favors the transcription of proapoptotic client genes, such as *DR5* (TRAIL Receptor-2); tribbles-related protein 3 (*TRB3*); BH3-only proteins of the Bcl-2 family *Bim*, *Bak*, *Bax*, *PUMA*, and *NOXA*; and the downregulation of anti-apoptotic genes of the same family (Figure 3). Another CHOP client gene is *GADD34*, which encodes for a phosphatase, is able to dephosphorylate eIF2α, and thus, overcome PERK-induced transcriptional block, further promoting the synthesis of pro-apoptotic proteins [59,60]. Furthermore, CHOP induces the expression of Ero1α, which promotes cell death through the hyperoxidation of ER proteins. The increased production of ROS induces a shift in the balance between reduced and oxidized residues of the cysteine domains of IP3R and this event favors calcium-dependent apoptosis [61].

Overall, the activation of transcription factors, kinase-dependent signaling pathways, and the regulation of members of the Bcl-2 family leads to activation of initiators caspases 8 and 9, and execution caspases 3, 6, 7, and 12. Among these, caspase 12 begins the final execution phase, even if its activation mechanisms are not completely understood [50,62].

In the context of cancer, some of the key components of the UPR signaling are up-regulated and chronically activate these adaptive mechanisms, thus promoting tumor progression and survival [63]. In such a view, new evidence connects the UPR with specific hallmarks of cancer, postulating new possible regulatory pathways, and suggests that this adaptive pathway may provide a mechanism of control of specific cancer functions, as capacity to adapt to hostile environments, escape apoptosis, and anticancer agents and reprogram cell metabolism [4].

### 2.3. The Role of the Inflammatory Signaling Cascade during the UPR

Emerging evidences suggest that there are points of connection between the UPR and the inflammatory cascade [52]. Indeed, ER stress induces inflammatory signaling and modulates nuclear factor-κB (NF-κB) activity [64], the principal transcriptional regulator of pro-inflammatory pathways [64]. In normal conditions, NF-κB is in an inactive status through binding with its constitutively expressed inhibitor, IκBα. Multiple cellular pathways activate IκBα kinase (IKK), which phosphorylates IκBα [64], leading to its proteasome degradation and consequent release and activation of NF-κB [64]. Thus, stress stimuli activate NF-κB nuclear translocation and the downstream upregulation of its inflammatory target genes [64] (Figure 4). In such a context, several genes regulated by NF-κB primarily promote survival, making NF-κB a key player in the development of invasive tumors, metastases, and resistance to several chemotherapeutic agents [65]. IRE1 is the key molecule responsible for the integration between UPR signaling and inflammatory response; during ER stress, the complex TRAF2/IRE1 is responsible for activation of NF-κB, as reported by Hu et al. [66] (Figure 4). Indeed, both NF-κB activity and IκBα degradation depend on IRE1 and are down-regulated in IRE1α-deficient cells, even though the exact mechanism used by IRE1α to regulate IKK activity is still unclear. In such a context, TRAF2 can also recruit and activate the pro-inflammatory pathway mediated by JNK and AP1 [67]. Altogether, this evidence supports the concept that ER signaling regulates important physiological or pathological processes and is responsible for the subtle balance between cell survival and death through the modulation of autophagy and bioenergetic and biosynthetic pathways [68].

## 3. Endoplasmic Reticulum Stress and UPR in Breast Cancer and Their Involvement in Drug Resistance

Breast cancer (BC) is the most common cancer in women and the second most common cause of cancer mortality; it has been estimated that almost 40,000 women die of breast cancer each year in USA [70]. It has been widely proposed that aberrant activation of UPR as well as the upregulation of UPR components are involved in BC progression and in resistance to apoptosis and drug therapy in BC cells [59].

### 3.1. Aberrant UPR Activation in Breast Cancer

The activation of the PERK-ATF4 axis is necessary for the progression of the breast tumors, having been demonstrated in both in vivo and in vitro BC cells models [71]. Indeed, the dysregulation of the PERK arm involves the activation of a series of signaling pathways that promote cell survival, and this occurs through the induction of a detoxifying action pathway, in which NRF2 plays a prominent role [72,73], or through the regulation of autophagy [74]. Consistently, the inhibition of PERK pathway enables the resensitization of BC cells to radiation [74]. However, this conclusion should be taken with caution since it has also been proposed that autophagy can be regulated by ATF4, independently of PERK [75], and that the activation of the autophagic pathway by PERK-ATF4 can also cause an imbalance towards apoptosis, transforming a pro-survival signal into a pro-death one [76]. Therefore, further studies are needed to clarify the exact molecular mechanism involved in the balance between pro-survival and pro-death signals linked to ER stress in breast carcinogenesis.

### 3.2. UPR Activation and Drug Resistance in Breast Cancer

Several studies evaluated the role of GRP78 and XBP1 as drivers of drug resistance, based on the general observation that they are upregulated in BCs compared with normal tissue [77,78]. In such a context, both GRP78 and XBP1 are involved in resistance to endocrine therapy, molecular-targeted agents, and traditional chemotherapeutics [59,79].

Luminal BC, the most common BC subtype (over 70% of all BCs), expresses estrogen receptor-α (ESR1; ERα) and is successfully treated with ERα-targeted therapies, which include receptor antagonists, such as tamoxifen or fulvestrant, and aromatase inhibitors that interfere in 17β-estradiol ligand production, such as anastrazole and letrozole [80]. Unfortunately, about half of ERα positive metastatic BCs respond to first line endocrine therapies, the remainders being de novo resistant [81,82], and many initially responsive tumors develop resistance to these agents [82,83]. Several mechanisms are responsible for resistance to endocrine therapy, as previously reviewed in References [5,82], and among them, the activation of the UPR pathway. In this context, a hypoxic tumor environment and glucose deprivation are involved in resistance to endocrine therapy, being responsible for a prolonged UPR activation [12,77,78], and GRP78 has a critical role as a regulator of endocrine responsiveness [9,79]. Mechanistically, estrogen induces the expression of GRP78 and XBP1 [78,84], both being UPR components overexpressed in, respectively, 60–70% and 80–90% human BCs [10,84] and GRP78 allows estrogen-dependent cells to survive in conditions of estrogen deprivation by binding and inhibiting the proapoptotic protein BCL-2 interacting killer (Bik) [13]. Furthermore, GRP78 overexpression prevents tamoxifen effectiveness, whereas its knockdown confers sensitivity to endocrine therapy [85,86]. Finally, Cook et al. demonstrated that GRP78 promotes acquired, but not de novo, resistance to tamoxifen in a rat model of mammary tumors [9].

UPR activation and XBP1 upregulation have been largely described in both ERα+ and ERα− BCs [82]. The crucial role of XBP1 in resistance to anti-estrogen therapy was initially predicted in an expression network study and confirmed using gene expression analysis on human BC specimens. Indeed, the UPR signature that includes XBP1 upregulation has been identified as a marker predictive of tamoxifen resistance in ER-positive BCs and has been associated with reduced time to recurrence (TTR) and poor survival [7]. In such a context, DNA microarray analysis clarified that XBP1 is an estrogen-responsive gene, expressed within the luminal cluster. The estrogen-dependent regulation of XBP1 expression was validated using different groups: XBP1 regions analysis identified ERα-regulated promoters, linking XBP1 transcription to ERα and estrogen signaling [87], and this was confirmed via chromatin immunoprecipitation experiments [88]. The transcriptional activity of ERα on XBP1 is associated to decreased sensitivity to endocrine therapy in tumors that express both proteins [6]. XBP1 has important growth functions: it regulates genes associated with cell cycle preventing cell cycle arrest and apoptosis, inhibits the mitochondrial apoptotic cascade and estrogen responsiveness in ER-α positive BC cells [89]. Finally, ERα signaling mediates a cytoprotective UPR in mitochondria in the presence of the accumulation of unfolded proteins in BC cells. The estrogen-independent activation of ERα induces a gene expression reprogramming with an increase of the proteasome activity and protection of organelles [90].

As previously mentioned, GRP78′s role in drug resistance is not restricted to endocrine therapy and to ER-positive BCs, being involved in resistance to anthracyclines [82,86] and being its expression elevated in different BC subtypes, such as HER2-like [91]. Indeed, the GRP78 protein level is also involved in the resistance to trastuzumab (Herceptin), a recombinant humanized monoclonal antibody that recognizes HER2 and is active in HER2-positive BCs [92]. The overexpression of HER2, primarily due to gene amplification, occurs in approximately 25–30% of invasive human BCs [93] and its downstream signaling pathway is responsible for several features of BC cells, i.e., cell proliferation and migration, resistance to apoptosis, activation of the angiogenic cascade, and the metastatic process [94]. Tastuzumab was approved by the FDA for the treatment of HER2-positive BCs in the adjuvant, neoadjuvant, and metastatic settings [95], based on the paradigm that the overexpression/amplification of the HER2 receptor represents a prognostic and predictive marker and a therapeutic target [94]. Proteomic profiling of trastuzumab-sensitive and resistant BC cells identified GRP78 among many putative mediators of drug resistance [94]. Furthermore, Kumandan et al. demonstrated the importance of the UPR as a potential mechanism to override the activity of trastuzumab and induce resistance through the activation of the PI3K/AKT axis and the overexpression of downstream oncogenes, such as Lipocalin 2 (*LCN2*) [96]. Indeed, ER stress and the attendant UPR represent a possible alternative way through which PI3K/AKT signaling is reactivated during HER2 inhibition by trastuzumab, ultimately leading to the upregulation of LCN2, hence potentially resulting in trastuzumab resistance.

Finally, the GRP78 protein level has important implications in the resistance to chemotherapeutics: breast tumors with GRP78 overexpression are indeed characterized by doxorubicin ineffectiveness [11]. Furthermore, GRP78 levels were found positively associated with shorter TTR in a cohort of BC patients treated with adriamycin-based chemotherapy and consequently proposed as a predictive factor of poor responsiveness to chemotherapy [11,86].

## 4. Molecular Chaperones in Protection from ER Stress and Drug Resistance in BC Cells

Molecular chaperones are responsible for correct protein folding in sub-cellular organelles and protection from ER stress and their upregulation represents an important adaptive mechanism frequently involved in human disease [1]. Indeed, cancer cells upregulate molecular chaperones to optimize protein synthesis and conjugate their increased metabolic and biosynthetic requirements with accelerated cell proliferation [14]. In such a view, molecular chaperones are key players in the intricate mechanism linking ER stress protection and metabolic rewiring to tumor progression and resistance to pharmacological agents [84]. It is well known that different molecular chaperones are overexpressed in breast cancer, i.e., Hsp90 and its mitochondrial homologues TRAP1, Hsp70, and Hsp27 [97]. Hsp90 and Hsp70 act as ATP-dependent chaperones, while Hsp27 acts as an ATP-independent chaperone [98].

### 4.1. TRAP1 (HSP75)

TRAP1 is a molecular chaperone with a prevalent mitochondrial localization and is selectively up-regulated in several human malignancies, including BC [99,100]. TRAP1 is characterized by antioxidant and antiapoptotic functions [101], and is involved in the protection against antiblastic agents, favoring a multidrug resistance phenotype [99,100]. Our group described a novel TRAP1 function outside mitochondria, at the interface between ER and mitochondria [99,102], where this chaperone interacts with the 19S proteasome subunit, 26S Proteasome AAA-ATPase Subunit RPT3 (TBP7), performing a quality control on a network of specific client proteins [102]. Noteworthy, this TRAP1 function is crucial for protein homeostasis, since TRAP1 attenuates global protein synthesis, whereas its silencing is associated with increased ubiquitination/degradation of nascent stress-protective client proteins [103]. Finally, this mechanism is responsible for cytoprotective functions and is highly conserved in human malignances supporting the concept that this pathway is relevant in drug resistance and tumor progression [88,89].

A mechanistic link was demonstrated between the upregulation of TRAP1 protein network, its capacity to modulate protein synthesis and prevent ER stress and resistance to paclitaxel and anthracyclins in human BCs [104,105]. Indeed, taxanes and anthracyclins are cytotoxic agents widely employed in the treatment of human BC [81] and are responsible for inducing ER stress in cancer cells [5,80]. TRAP1 cytoprotective activity toward these chemotherapeutics strongly relies on its capacity to protect itself from ER stress, being the molecular chaperone upregulated in about 50% of human BCs and co-expressed with the ER stress marker, GRP78 [104,105]. Mechanistically, ER-associated TRAP1 is responsible for quality control on 18 kDa Sorcin, a TRAP1 mitochondrial client protein involved in TRAP1 cytoprotective pathway, and consequently modulates mitochondrial apoptosis, and, thus favors resistance to paclitaxel and anthracyclins [100,104,105].

### 4.2. HSP90

Hsp90 is a molecular chaperone that assists other proteins to fold properly, and, among several others, HER2 was described as a Hsp90 client protein [106]. In addition, preclinical studies pointed the attention on a mutual regulation between HER2 and Hsp90 [107]. Thus, based on this evidence, Hsp90 has been widely studied as a molecular target in human HER2-positive BC cells [24,106]. However, it should be emphasized that the therapeutic effect of Hsp90 inhibitors as single agents or in combination with antibody directed against HER2 did not produce satisfactory results [24] and this is likely due to compensatory effects promoted by Hsp90 inhibition, promoting the up-regulation of Hsp70 and Hsp27 [24]. Furthermore, pre-clinical studies correlated Hsp90 overexpression with aggressive behavior [108], and with self-renewal in BC stem cells [109,110], suggesting the importance of this chaperone in the regulation of BC metabolism.

### 4.3. HSP27

Hsp27 (also called HspB1) is a small heat shock protein, mainly involved in protein folding and upregulated in cells exposed to stress conditions [111]. Several studies linked Hsp27 to ER stress [112], activation of autophagy [113,114], and protection from apoptosis and drug resistance [115], based on the observation that the molecular chaperone is overexpressed in many human malignancies, including BCs [116]. Zhang D. et al. reported that Hsp27 is specifically upregulated in HER2-positive breast tumors [117], and different authors found that the upregulation of Hsp27 in BC cells reduces trastuzumab sensitivity by increasing HER2 protein stability [118] and reduces doxorubicin susceptibility by inhibiting the apoptotic cascade [119]. Consistent with a mutual cooperation between molecular chaperones, Lee C-H et al. reported that the inhibition of Hsp27 potentiates the activity of Hsp90 inhibitors in BC stem-like cells [120].

## 5. Future Prospective: Autophagy and UPR as Novel Molecular Targets

The evolution in the knowledge of ER functions allowed for the identification of numerous roles for ER in physiology and pathology. The deeper understanding of this organelle and its homeostatic regulators, the UPR response, provided relevant opportunities to exploit this information on multiple fronts. The UPR integrates different cellular processes from protein folding to the control of transcription and translation, from protein degradation to the regulation of signaling pathways responsible for cell fate. Chronic activation of UPR is a general feature of cancer cells and it is considered a mechanism of tumor progression and resistance to apoptosis and anticancer agents. In such a perspective, ER stress regulators are presently considered novel molecular biomarkers of prognosis, as well as potential targets for therapeutic interventions. In BC, the treatment with ERα targeting therapies and cytotoxic agents induces the activation of pro-survival UPR pathways conferring intrinsic resistance to endocrine and molecular-targeted agents and chemotherapeutics through a mechanism driven by GPR78 and XBP1 upregulation. In such a scenario, a possible pharmaceutical strategy to counteract tumor progression and drug resistance is to target UPR pro-survival components and the complex interplay between UPR, autophagy, and apoptosis, thus favoring a shift toward cell death instead of cell survival. Indeed, Cook et al. explored GRP78 and autophagy as potential targets to overcome endocrine responsiveness in ERα-positive BC cells. They observed that GRP78 integrates signaling pathways involving UPR and autophagy and that the simultaneous knockdown of GRP78 and beclin-1 re-establishes sensitivity to antiestrogen therapy in resistant cells [9]. Furthermore, GRP78 was investigated as a target to deliver cytotoxic peptides or suicide transgene in BC in vivo. A GRP78-binding peptide fused to a pro-apoptotic moiety (i.e., BMTP78) was evaluated to selectively kill GRP78-positive BC cells. Indeed, BMTP78 was shown to inhibit primary tumor growth and the metastatic cascade in preclinical BC models [121]. Other authors used GRP78 overexpression in inflammatory BCs to deliver the human herpes simplex virus thymidine kinase type-1 (HSVtk) transgene. Interestingly, GRP78-targeting by adeno-associated virus/phage (AAVP) particles resulted in cytotoxic activity toward tumor cells by selective activation of the prodrug ganciclovir at tumor sites [122].

In recent years, autophagy has been proposed as an innovative target for anticancer interventions with the introduction of autophagy inhibitors as potential anticancer drugs [82]. Multiple agents are active inhibitors of different phases of autophagy, but the only clinically-approved autophagy inhibitor is an anti-malarial chloroquine and its derivatives, such as hydroxychloroquine (HCQ) [123]. HCQ inhibits lysosomal acidification and prevents the degradation of autophagosomes, thereby suppressing autophagy. HCQ treatment induces a prolonged growth arrest in vitro, and preclinical studies showed that the inhibition of autophagy by HCQ is able to overcome chemotherapeutic resistance in several tumor cell lines and animal cancer models [124]. In ERα-positive BC cell lines, combination therapy with HCQ and tamoxifen showed superior activity compared to endocrine monotherapy [125]. Consistently, HCQ showed the capacity to potentiate the anticancer activity of bortezomib, temozolomide, temsirolimus, and doxorubicin in several cancer models, and an HCQ single agent showed clinical activity in patients with melanoma, colorectal cancer, myeloma, and renal cell carcinoma [123]. While this evidence represents the proof of concept that targeting autophagy may provide a valid anticancer strategy, future directions in this field need to identify (i) novel and more selective inhibitors of autophagy or UPR transducers, (ii) subsets of patients that are likely susceptible to autophagy inhibition, and (iii) novel predictive biomarkers to select these tumors for clinical interventions.

## Figures and Tables

**Figure 1 ijms-20-00857-f001:**
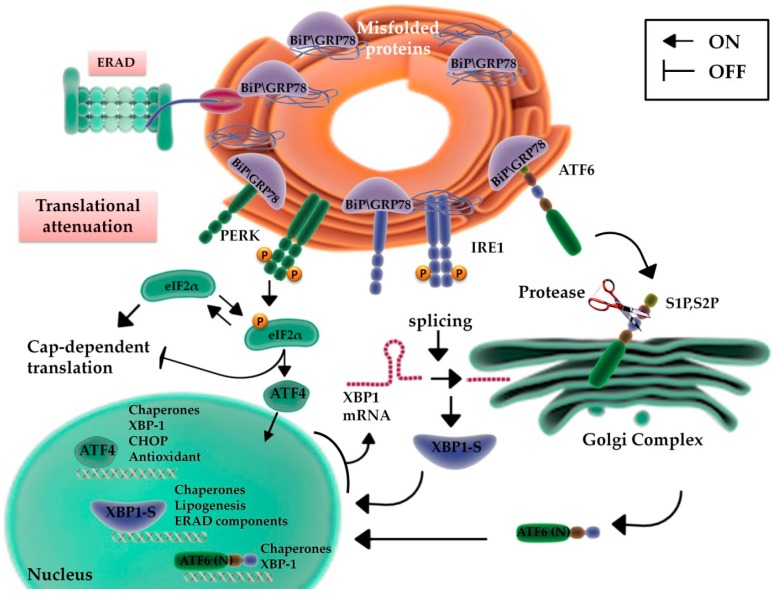
Unfolded protein response activation during Endoplasmic Reticulum (ER) stress conditions. Accumulation of misfolded proteins in endoplasmic reticulum induces the activation of three ER stress sensors: the activating transcription factor 6 (ATF6), the inositol requiring enzyme 1 (IRE1), and the protein kinase RNA-activated (PKR)-like ER kinase (PERK). Under physiological conditions, their activation is prevented by binding to BiP/GRP78. When misfolded proteins increase in the ER, GRP78 is recruited for protein folding and releases Unfolded protein response (UPR) sensors. Consequently, ATF6 is transported to the Golgi and activated, following cleavage by Site-1 protease (S1P) and Site-2 protease (S2P). Cleaved ATF6 (ATF6(N)) behaves as a transcription factor, inducing the expression of ER chaperones and the transcription factor, XBP1. Activated IRE1 induces the splicing of XBP1 mRNA, and the resulting spliced XBP1 protein (XBP1s) translocates to the nucleus and regulates the transcription of genes involved in ER-associated degradation (ERAD). PERK, when activated, hampers general protein synthesis via the phosphorylation of eIF2α and enables the translation of ATF4. Under physiological conditions, damaged or incorrectly folded proteins are sequestered and eliminated through endoplasmic reticulum-associated protein degradation (ERAD).

**Figure 2 ijms-20-00857-f002:**
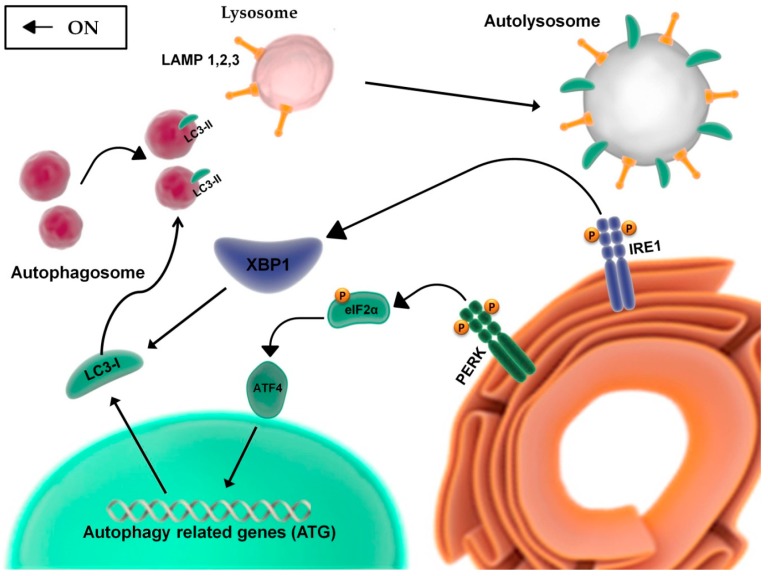
Crosstalk between UPR and autophagy. The autophagic process involves a molecular crosstalk with UPR transducers: (i) PERK/eIF2α pathway induces the expression of autophagy-related genes through ATF4; and (ii) the activation of XBP1, downstream to IRE1, favors the recruitment of soluble LC3-I to the membranous structures, giving rise to the autophagosome, and to its transformation into LC3-II, the membrane-associated form of LC3-I. The autophagosome complex, conjugated with LC3-II, captures organelles or molecules destined for degradation and merges with lysosome-expressing integral membrane proteins, LAMPs (1,2,3), and this leads to the formation of the autolysosome complex.

**Figure 3 ijms-20-00857-f003:**
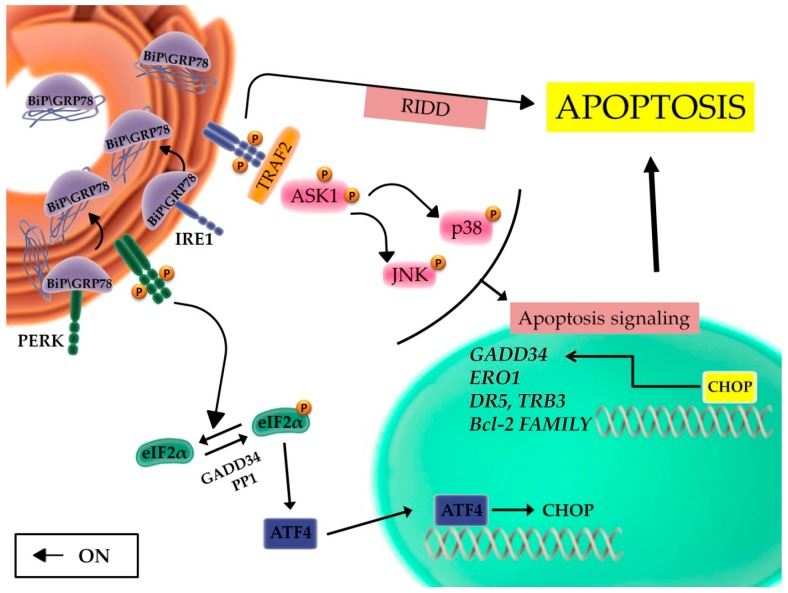
Apoptotic signaling during ER stress. Prolonged or severe ER stress conditions induce apoptotic signaling through the activation of IRE1 and PERK pathways. Upon activation of the IRE1 pathway, IRE1 recruits TRAF2 and ASK1 on the ER membrane and this induces the apoptotic response upon modulation of the balance between proapoptotic and antiapoptotic genes. Furthermore, IRE1 digests specific miRNAs through its endoribonuclease domain (IRE1-dependent decay of mRNA, RIDD). PERK pathway activates ATF4, which in turn enhances the transcription of CHOP, a transcription factor responsible for the regulation of many proapoptotic genes (i.e., *GADD34*, *ERO1*, *DR5*, *TRB3*, *Bcl-2* family genes). In this complex scenario, GADD34 and PP1 promote the dephosphorylation of eIF2α, leading to enhanced protein synthesis and the increase of protein load into the ER.

**Figure 4 ijms-20-00857-f004:**
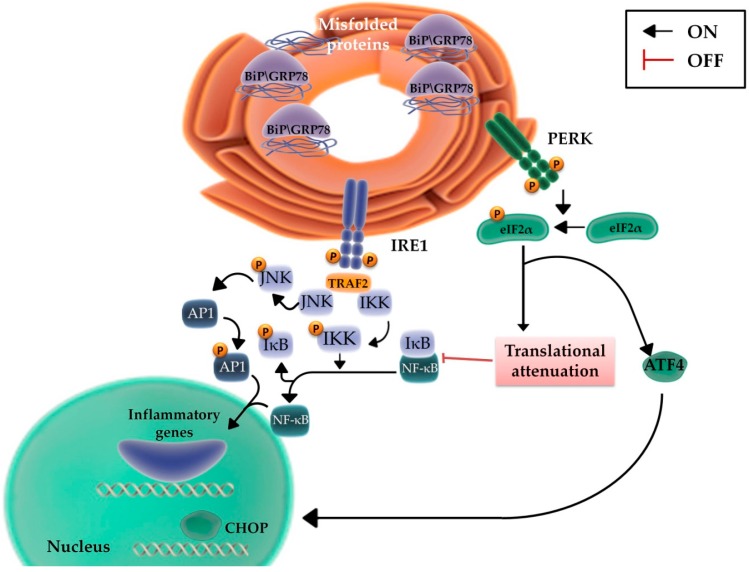
UPR-associated inflammatory signaling pathways. The activation of NF-κB requires the phosphorylation of its inhibitor, IκBα, via IKK, leading to IκBα proteasome degradation and the consequent release of NF-κB in its active form. During ER stress, activated IRE1 forms a complex with TRAF2 and activates IKK, which in turn induces IκB degradation, the subsequent activation of NF-κB and the transcription of pro-inflammatory genes. TRAF2 also induces the phosphorylation of JNK and the up-regulation of other pro-inflammatory genes through activated AP1. Furthermore, activated PERK promotes NF-κB activation via translational attenuation of IκB [69].

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
