# Peer review of "Endoplasmic Reticulum Stress and Unfolded Protein Response in Breast Cancer: The Balance between Apoptosis and Autophagy and Its Role in Drug Resistance"

_ijms, 2019, doi:10.3390/ijms20040857_

Round 1
Reviewer 1 Report
In this review, Sisinni and colleagues analyze the relationship between endoplasmic reticulum (ER) stress, apoptosis and autophagy in physiological and pathological contexts. The final aim is to understand such inter-connections in human breast cancer (BC) and in resistance to cancer therapies. There are already a lot of reviews on ER stress and Cancer, but the question of the role of ER stress in drug resistance is original.
The review is well written and the english language and style are fine. The work is well organized with a first part (most important one) describing ER function, then a second part focusing on ER stress in BC and in drug resistance. The review ends with a perspective paragraphe.
Major Comments:
- The title of the review is not in adequacy with the content, because most of the review describe ER stress in a general way. I suggest the authors to focus on ER stress and Breast Cancer, by reducing/summarizing the first part (keeping references on cancer), and by enriching the second part.
- The Figure 1 is useless under this form. I suggest the authors to remove it or to improve it.
- Regarding the second part (ER stress in BC and in drug resistance), more than 1/3 of the text is focusing on the molecular chaperone TRAP1, that is a protein described by the authors of the review. I suggest the authors to summarize their findings, and to add other examples of chaperones that may be also interesting to consider as potential therapeutic targets.
- Considering the last paragraph ("Future prospective"), the discussion on UPR as novel molecular targets is too descriptive and needs to be reconsidered by authors.
Minor Comments:
- Page 3, line 41-43: The synthesis of cap-independent transcripts is present in the text but not in the figure. The authors have to add it on the figure. May the authors also verify that the corresponding reference is the good one (ref 24).
- Page 4, line 1-3: A reference is missing ("...and therapy resistance (ref).").
- Figure 3: an arrow is missing (linking XBP1 to LC3-I).
- Page 5, line 2: error mitocondria >> mitochondria
- Page 5: I suggest the authors to find a better reference than 41 for the description of the intrinsic apoptotic pathway.
- Page 5: May the authors check and validate following references 47/52/53.
- Figure 5: It is known in the litterature that TRAF2 can also lead to JNK phosphorylation and AP1 translocation to the nucleus, leading to the transcription of pro-inflammatory cytokines. This pathway is missing on the figure (and in the text), I suggest the author to include it.
- Page 7, line 31-32 (legend Figure 5): missing reference ("...attenuation of IkB (ref).").
- Page 8, line 41: the reference of "Kumandan et all." is missing (in the text and in the References part).
- The references cited in the text as "xxxx et all" have to be replaced by "xxxx et al".
Author Response
Referee #1
In this review, Sisinni and colleagues analyze the relationship between endoplasmic reticulum (ER) stress, apoptosis and autophagy in physiological and pathological contexts. The final aim is to understand such inter-connections in human breast cancer (BC) and in resistance to cancer therapies. There are already a lot of reviews on ER stress and Cancer, but the question of the role of ER stress in drug resistance is original.
The review is well written and the English language and style are fine. The work is well organized with a first part (most important one) describing ER function, then a second part focusing on ER stress in BC and in drug resistance. The review ends with a perspective paragraph.
Major Comments:
- The title of the review is not in adequacy with the content, because most of the review describe ER stress in a general way. I suggest the authors to focus on ER stress and Breast Cancer, by reducing/summarizing the first part (keeping references on cancer), and by enriching the second part.
Reply: The second part of the Review was enriched by discussing the role of UPR in breast cancer progression. Indeed, a new paragraph was added to highlight the role of aberrant URP activation in human breast carcinoma. The paragraph on ER stress and drug resistance was reorganized. We suggest that, with these revisions, the Review is more complete and adequate with the Title. Regarding the request to summarize the first part of the Review, this was done in part. Indeed, we suggest that the description of the molecular mechanisms/pathways involved in ER stress response is important for a better understanding of the second part of the Review, especially for readers not deeply involved in this field.
- The Figure 1 is useless under this form. I suggest the authors to remove it or to improve it.
Reply: Figure 1 was removed, as requested.
- Regarding the second part (ER stress in BC and in drug resistance), more than 1/3 of the text is focusing on the molecular chaperone TRAP1, that is a protein described by the authors of the review. I suggest the authors to summarize their findings, and to add other examples of chaperones that may be also interesting to consider as potential therapeutic targets.
Reply: Two additional examples of molecular chaperones involved in the relationship between ER stress and drug resistance in breast cancer were added (i.e., HSP90 and HSP27). HSP90 was discussed based on the rationale that HER2 is among its client proteins. HSP27 was selected based also on the request of Referee #2. TRAP1 role in ER stress and drug resistance was also summarized.
- Considering the last paragraph ("Future prospective"), the discussion on UPR as novel molecular targets is too descriptive and needs to be reconsidered by authors.
Reply: The concept that UPR may represent a novel molecular target was discussed deeply. Three additional studies were discussed providing the rationale that GRP78 can be a target either to restore endocrine responsiveness or to deliver cytotoxic peptides or suicide transgene in BC in vivo.
Minor Comments:
- Page 3, line 41-43: The synthesis of cap-independent transcripts is present in the text but not in the figure. The authors have to add it on the figure. May the authors also verify that the corresponding reference is the good one (ref 24).
- Page 4, line 1-3: A reference is missing ("...and therapy resistance (ref).").
- Figure 3: an arrow is missing (linking XBP1 to LC3-I).
- Page 5, line 2: error mitocondria >> mitochondria
- Page 5: I suggest the authors to find a better reference than 41 for the description of the intrinsic apoptotic pathway.
- Page 5: May the authors check and validate following references 47/52/53.
- Figure 5: It is known in the litterature that TRAF2 can also lead to JNK phosphorylation and AP1 translocation to the nucleus, leading to the transcription of pro-inflammatory cytokines. This pathway is missing on the figure (and in the text), I suggest the author to include it.
- Page 7, line 31-32 (legend Figure 5): missing reference ("...attenuation of IkB (ref).").
- Page 8, line 41: the reference of "Kumandan et all." is missing (in the text and in the References part).
- The references cited in the text as "xxxx et all" have to be replaced by "xxxx et al".
Reply: All minor comments were addressed, as requested by the Referee.
Reviewer 2 Report
The authors clearly reviewed the signaling pathway of unfold protein responses in ER stress, autophagy, apoptosis and breast cancer drug resistance. The manuscript could be interested to researchers in the field of cell biology and cancer biology. Some minor revisions should be considered.
It is not clear why use DNA structure in Figure 1 and should be considered to use another better presentation way to express.
Hsp27 has been linked to autophagy, apoptosis, ER stress, and drug resistance in cancers including breast cancer (PMID: 15864808, 22445681, 22675041, 25962073, 29062135, 29592877, etc.). The authors should consider to add a section to discuss the role of Hsp27 in this review.
Author Response
Referee #2
The authors clearly reviewed the signaling pathway of unfold protein responses in ER stress, autophagy, apoptosis and breast cancer drug resistance. The manuscript could be interested to researchers in the field of cell biology and cancer biology. Some minor revisions should be considered.
- It is not clear why use DNA structure in Figure 1 and should be considered to use another better presentation way to express.
Reply: As suggested by the Referee #1, Figure 1 is useless. Thus, it was removed.
- Hsp27 has been linked to autophagy, apoptosis, ER stress, and drug resistance in cancers including breast cancer (PMID: 15864808, 22445681, 22675041, 25962073, 29062135, 29592877, etc.). The authors should consider to add a section to discuss the role of Hsp27 in this review.
Reply: A paragraph was dedicated to HSP27 in the second part of the Review, highlighting the connection between HSP27 and autophagy, apoptosis, ER stress, and drug resistance. To fulfill the request of the Referee #1, HSP27 and HSP90 were discussed as additional examples of molecular chaperones linker to ER stress and drug resistance in breast carcinoma.